# Predicting Assembly Geometric Errors Based on Transformer Neural Networks

Wu Wang [1,2], Hua Li [1], Pei Liu [3], Botong Niu [2], Jing Sun [2] and Boge Wen [3,*]

[1] School of Computer Science and Technology, Changchun University of Science and Technology, Changchun 130022, China; wxiongxiong@163.com (W.W.); lihua@cust.edu.cn (H.L.)
[2] North Navigation Control Technology Co., Ltd., Beijing 100176, China; niubotong1994@163.com (B.N.); sunjing1777@126.com (J.S.)
[3] Collage of Computer Science and Engineering, Changchun University of Technology, Changchun 130012, China; liu_pei0077@126.com
* Correspondence: wenboge@ccut.edu.cn

**Abstract:** Using optimal assembly relationships, companies can enhance product quality without significantly increasing production costs. However, predicting Assembly Geometric Errors presents a challenging real-world problem in the manufacturing domain. To address this challenge, this paper introduces a highly efficient Transformer-based neural network model known as Predicting Assembly Geometric Errors based on Transformer (PAGEformer). This model accurately captures long-range assembly relationships and predicts final assembly errors. The proposed model incorporates two unique features: firstly, an enhanced self-attention mechanism to more effectively handle long-range dependencies, and secondly, the generation of positional information regarding gaps and fillings to better capture assembly relationships. This paper collected actual assembly data for folding rudder blades for unmanned aerial vehicles and established a Mechanical Assembly Relationship Dataset (MARD) for a comparative study. To further illustrate PAGEformer performance, we conducted extensive testing on a large-scale dataset and performed ablation experiments. The experimental results demonstrated a 15.3% improvement in PAGEformer accuracy compared to ARIMA on the MARD. On the ETH, Weather, and ECL open datasets, PAGEformer accuracy increased by 15.17%, 17.17%, and 9.5%, respectively, compared to the mainstream neural network models.

**Keywords:** long sequence forecasting; geometric errors; assembly precision; artificial intelligence; data processing



## 1. Introduction

Primary manufacturing processes encompass component assembly, product design, and component machining. This stage is also the most challenging to standardize and automate. The integration of intelligence with industrialization has become a focal point of recent advancements with the development of information technology. A vast amount of data has accumulated in the industrial sector with the adoption of intelligent devices. Predicting the quality of assembly processes has become possible through the analysis of historical industrial big data [1–4]. For example, the assembly process for folding rudder blades for unmanned aerial vehicles entails selecting components from a pool of qualified parts and assembling them to minimize the deviation of the rudder oscillation. All components within a permissible tolerance range are considered normal and qualified products.

However, this assembly method exhibits drawbacks, including significant variations in assembly accuracy and inconsistent inspection indicators. In practical scenarios, these issues may arise from excessive adjustments in the assembly process, resulting in non-compliance with inspection indicators. Consequently, this reduces assembly efficiency and increases the time cost due to incorrect assembly. To mitigate these effects, data-driven approaches can analyze and discover patterns within the vast amount of data

generated during production and manufacturing processes. This enables the improvement of product quality by precisely controlling assembly accuracy. Currently, this method is the most feasible.

In order to improve assembly geometric errors, the majority of manufacturing plants undergo digital transformation [5–7]. Utilizing traditional data-driven methods involves processing and predicting collected data, thereby establishing a mapping between the physical world and the digital world, known as a digital twin. Although these traditional data-driven methods exhibit strong operability (such as utilizing genetic algorithms to analyze assembly or disassembly sequences, among others) and can effectively improve overall assembly geometric errors, their accuracy and robustness are compromised when predicting errors in the assembly process of increasingly complex products [8–11]. To overcome the limitations of traditional data-driven methods, machine learning techniques have emerged as the mainstay in this field. However, existing research employing machine learning methods often develops algorithms only for specific combinations of components, resulting in poor generality of datasets for other models. Moreover, these methods are sensitive to environmental factors, rendering many Assembly Geometric Error accuracy predicting algorithms ineffective if environmental changes occur during data collection [12–14].

Previous studies have shown that using Transformers as a basis has yielded good results in prediction tasks such as sensor network monitoring [15], human behavior prediction [16], energy and smart grid management [17], economics and finance [18], and disease propagation analysis [19]. However, these methods are developed for time series forecasting, emphasizing the impact of continuous changes in datasets. In contrast, predicting assembly geometric errors requires capturing implicit information between assembly components. Therefore, using neural networks based on time series data directly for predicting assembly geometric errors may not yield satisfactory results.

To address the issue of predicting assembly geometric errors not related to time series using Transformers, this paper analyzed the data generated during the production manufacturing process and input paired data with assembly relationships into the neural network structure, facilitating a more robust establishment of correlations between the data. Consequently, the paper enhanced the long time series network structure and introduced a self-attention network structure based on assembly data pairs to achieve more accurate predictions of overall assembly geometric errors. To better evaluate the testing results, we acquired actual assembly data for folding rudders in unmanned aerial vehicles to establish a dataset of measurement data and errors for training and testing purposes.

In summary, this paper proposes a novel pair feature distance method to enhance data correlation for predicting assembly geometric errors. Furthermore, based on this data pair, a self-attention structure is introduced, focusing on global features and extracting local features.

The subsequent sections of this article are organized as follows: Section 2 reviews current research related to assembly data errors. Section 3 introduces the overall structure of neural networks, detailing the processing of pair feature distances in assembly data and the internal composition of encoders and decoders. Section 4 presents and analyzes the experimentally obtained predicted results. Finally, Section 5 provides a comprehensive summary of the entire article.

## 2. Background

The accuracy control prediction system in the assembly field is predominantly propelled by artificial-intelligence-based neural networks. Earlier research primarily concentrated on artificial neural network systems, grey prediction systems, fuzzy control theory, and various other aspects. Examples encompass numerical algorithms analyzing the final unfolding angles of adjusted joints in a linkage, neural networks predicting final errors for achieving higher accuracy, and adaptive support vector machines (ASVMs) based on the SVM framework predicting the assembly quality of automotive sunroofs. Alternative approaches include using artificial intelligence techniques to narrow down the search space

for assembly sequence planning, considering an analysis of the limitations of existing methods [20–22]. Leveraging the potent fitting capability of machine learning, these methods can yield superior results compared to traditional algorithms. Another category of methods addresses scenarios in which traditional algorithms are insufficient for analyzing complex mechanical products. In such scenarios, artificial intelligence can be trained and fitted based on data to predict the performance of assembling complex products, for example, the Assembly Quality Adaptive Control System (A_QACS). This system is proposed for assembly recognition of complex mechanical products under uncertainty. Additionally, a proposed multi-objective discrete particle swarm optimization algorithm is designed to enhance the efficiency of assembly planning [23,24].

In recent years, artificial-intelligence-based neural networks have rapidly advanced, particularly with the introduction of the Transformer architecture based on the self-attention mechanism. Consequently, methods developed based on the Transformer approach have gradually replaced sequential or time-related data analysis and prediction techniques [25–27]. Therefore, this paper aims to propose a new approach for predicting assembly errors based on a neural network architecture using the Transformer framework.

## 3. Model Architecture

This article analyzes the data characteristics of Assembly Geometric Errors to establish a structure named Assembly Geometric Error Embeddings (AGEE), which associates assembly relationships as input. To better handle structures with assembly relationships and the final errors caused by errors in other parts, this article introduces an efficient neural network architecture named PAGEformer. PAGEformer perceptively considers both long sequence relations and global consideration of local information. This architecture comprises an Encoder structure and a Decoder structure. The Encoder transforms the input sequence into hidden representations or feature vectors [28]. It consists of multiple identical layers, each containing prob attention, conv1d, and layer norm. The prob attention structure filters weights representing global information and local assembly relationship information through KL divergence computation, enabling more accurate error prediction. The purpose of the Decoder is to generate the target sequence, including prob attention, conv1d, layer norm, and a regular multihead attention structure. The Decoder incorporates the position of the data to be predicted into the AGEE structure, yielding the final prediction results. An overview is presented in Figure 1.

### 3.1. Assembly Geometric Error Embeddings

In typical scenarios, input data undergo differentiation, incorporating weighted positional information to capture temporal sequences. Nevertheless, this method might not comprehensively depict the assembly relationships among diverse components. In response to this limitation, our paper introduces the Assembly Geometric Error Embeddings method. This method establishes local assembly relationships among components and long-term sequential connections in the data before entering the Encoder and Decoder. This methodology aids the Encoder and Decoder in effectively capturing feature information, thereby augmenting the accuracy of prediction errors.Consider the input data as $X = \{x_1, \cdots, x_i | x_i \in d_N\}$. Classify the data into two categories, namely "gap" and "fill", based on the assembly relationship. Next, vectorize the data and perform calculations using the provided formula in Equation (1).

$$Y(j) = \sum_{i=0}^{N-1} X(j+i)W(i) + b \tag{1}$$

where $X(j+i)$ represents the $(j+i)$-th element of the input sequence, $W(i)$ denotes the $i$-th weight of the convolutional kernel, $b$ is the bias term, and $N$ represents the input length. By varying the number of convolutional kernels, we can derive distinct output dimensions denoted as $Q \in \mathbb{R}^{L_Q \times d}$, $K \in \mathbb{R}^{L_K \times d}$, and $V \in \mathbb{R}^{L_V \times d}$. Specifically, $LQ$ represents

the dimension formed by stacking the outcomes of $Q$ Conv1D convolutions, $LK$ is the dimension formed by stacking the outcomes of Conv1D convolutions, and $LV$ is the dimension formed by stacking the outcomes of $V$ Conv1D convolutions.

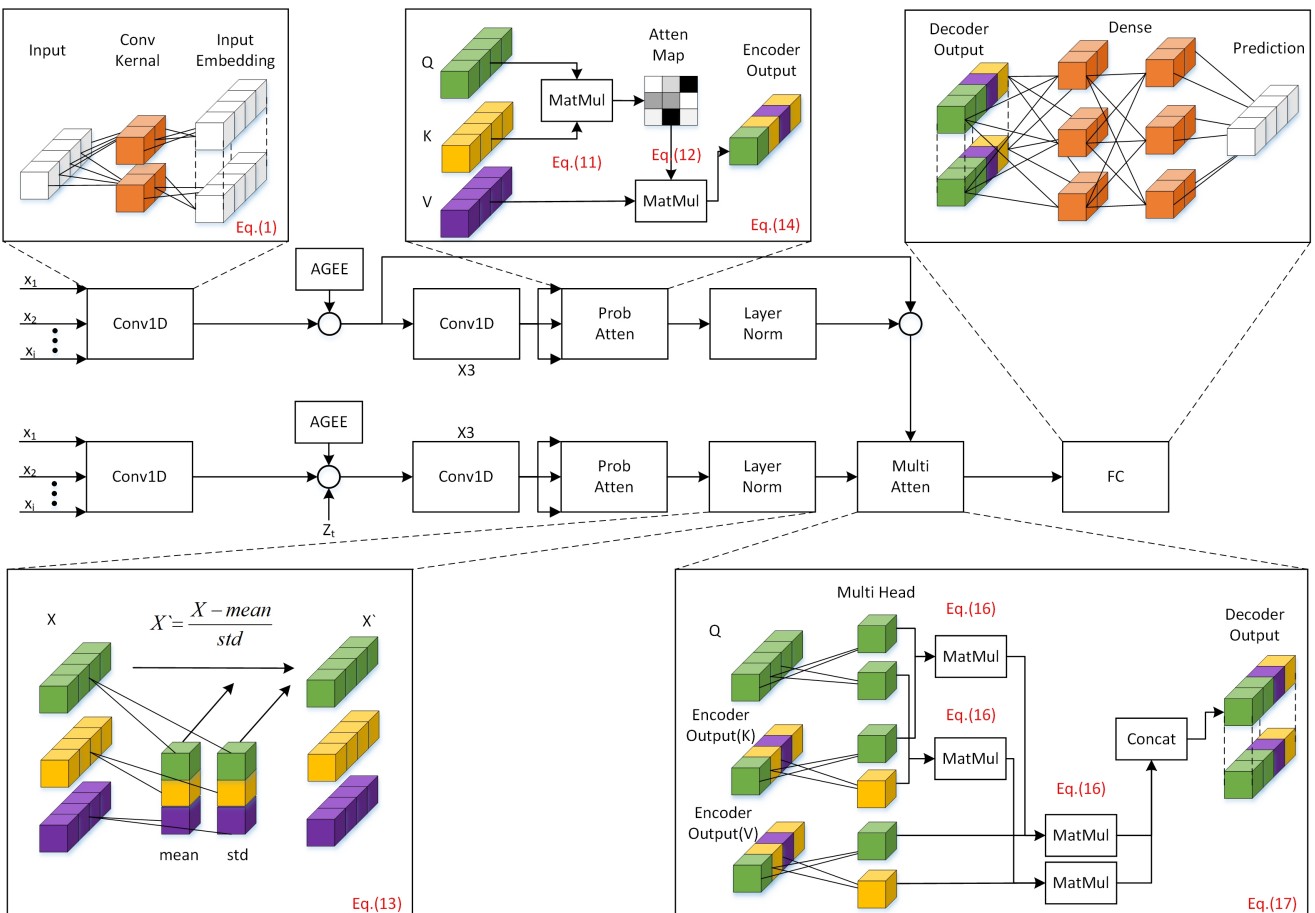

**Figure 1.** Illustration of the structure of PAGEformer, elucidating how input data are overlaid with assembly relationship information through AGEE. The Encoder comprises prob attention, conv1d, and layer norm. The Decoder, in addition to receiving the output from the Encoder through multihead attention, shares a similar structure with the Encoder. Finally, the fully connected layer is utilized to generate the ultimate prediction result. Here, conv1d is used to increase the dimensionality of the data and is combined with prob attention. Prob attention aims to better capture the correlations between Assembly Geometric Errors. Conventional self-attention is primarily used to establish features between long time-series data. Therefore, we processes the data pairs in AGEE to facilitate prob attention in extracting the relationships between assembly data. Additionally, prob attention filters out Q values that better reflect assembly relationships through KL divergence. For detailed calculation methods, please refer to Section 3.2. Layer norm is a common standardization method that helps the model converge better during training. Multi Atten is a common self-attention mechanism used to enhance the correlations between data. FC is used to project high-dimensional information onto the prediction dimension.

In this paper, the fitting relationships for shaft and slider are classified as "gap", while those for holes and slots are categorized as "fill". The positions corresponding to "gap" and "fill" are determined using Equations (2) and (3).

$$PE_{(pos,2i)} = sin(pos/(10000)^{2i/d_{model}}) \tag{2}$$

$$PE_{(pos,2i+1)} = cos(pos/(10000)^{2i/d_{model}}) \tag{3}$$

where $d_{model}$ denotes the model's dimension, and $i \in \{1, \cdots, d_{model}/2\}$. The embedding position is represented by *PE*, with dimensions identical to those of the data for filling and gaps, pos signifies the position of filling or gaps, *d* represents the dimension of *PE*, $2i$ represents even dimensions, and $2i + 1$ signifies odd dimensions.

Let us assume the filling data input for a set of *t* components is represented by $F^t = \{f_1^t, \cdots, f_{Lx}^t | f_i^t \in d_{Lx}\}$, and the gap data input is denoted as $G^t = \{g_1^t, \cdots, g_{Lx}^t | g_i^t \in a_{Lx}\}$. By evaluating Equation (1), $F^t$, and $G^t$ as follows: $F^t = \left\{ f_1^t, \cdots, f_{Lx}^t \mid f_i^t \in^{d_{Lx} \times d_{model}} \right\}$ and gap data $G^t = \left\{ g_1^t, \cdots, g_{Lx}^t \mid g_i^t \in^{d_{Lx} \times d_{model}} \right\}$. Combining Equation (2) and Equation (3), the new vector containing "fill" and "gap" information can be computed according to Equations (4) and (5).

$$\overline{f_i^t} = f_i^t + PE_{(pos, L_x \times (t-1) + i)} \tag{4}$$

$$\overline{g_i^t} = g_i^t + PE_{(pos, L_x \times (t-1) + i)} \tag{5}$$

where $i \in 1, \cdots, L_x$. The two input vectors are concatenated, which can be expressed by Equation (6).

$$INPUT^E = Concat(\overline{F^t} + \overline{G^t}) \in (Lx + lx) \times d_{model} \tag{6}$$

where $\overline{F^t} \in^{L_x \times d_{model}}$ and $\overline{G^t} \in^{L_x \times d_{model}}$. The obtained final vectors serve as the input vectors for the neural network. AGEE as depicted in Figure 2.

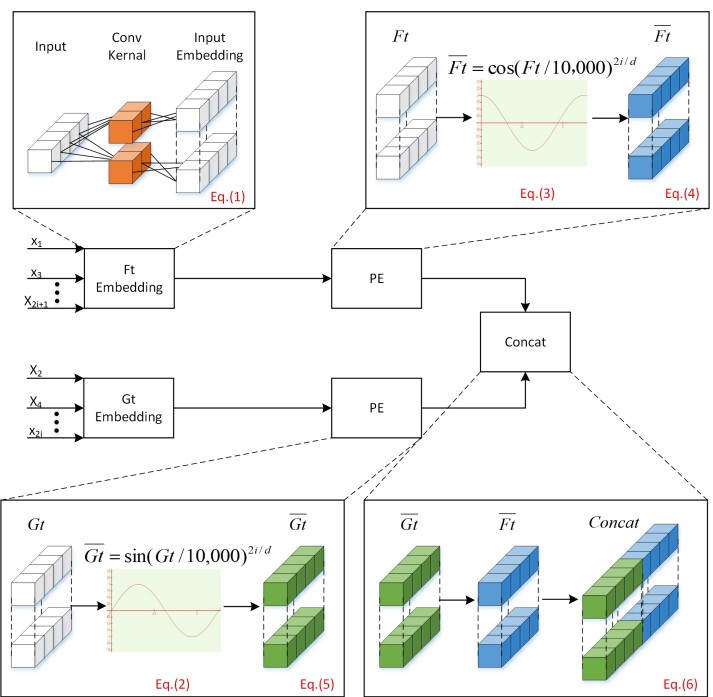

**Figure 2.** Segregation of the input into "gap" and "fill" components. Each component is subsequently subjected to conv1D transformation, resulting in the Lx×dmodel dimension. Following this, each component is individually combined with PE, and ultimately, the outcomes of both are concatenated to form INPUTE.

### 3.2. Encoder

The Encoder consists of prob attention, conv1d, and layer normalization components. In the attention mechanism, there might be partially activated Queries, permitting specific Query key dot product calculations with minimal contribution to the prediction to overpower the probability distribution after the softmax function. This suggests that certain Queries result in inefficient utilization of computational resources. To mitigate this limitation and improve prediction accuracy, it is imperative to filter out fully activated Queries. Non-activated Queries display characteristics akin to a uniform distribution;

therefore, representative Queries can be identified by computing the Kullback–Leibler (KL) divergence between the distribution of distinct Queries and a uniform distribution. Attention can be expressed by Equation (7).

$$A(Q, K, V) = softmax(\frac{QK^T}{\sqrt{d}})V \tag{7}$$

where $Q \in \mathbb{R}^{L_Q \times d_{model}}$, $K \in \mathbb{R}^{L_K \times d_{model}}$, and $V \in \mathbb{R}^{L_V \times d_{model}}$. $Q$, $K$, and $V$ represent three distinct results derived from applying $INPUT^E$ to various conv1D operations, "$i$" represents the $i$-th row in $Q$, $V$, and $j$ represents the $j$-th column in $Q$, $K$, $V$. Let $q_i$, $k_i$, $v_i$ stand for the $i$-th row in $Q$, $K$, and $V$, respectively. $A(Q, K, V)$ represents the attention score of the $i$-th $q$ in the sequence, $p$ is the probability score of the $i$-th $q$ and $j$-th $k$ calculated from $f(q_i, k_j) = \exp\left(\frac{q_i k_j^T}{\sqrt{d}}\right)$. The prob_attention is expressed by Equations (8) and (9).

$$A(q_i, K, V) = \sum_j \frac{f(q_i, k_i)}{\sum_l f(q_i, k_i)} v_i \tag{8}$$

$$p(k_j \mid q_i) = \frac{f(q_i, k_j)}{\sum_i f(q_i, k_j)} \tag{9}$$

The probability of the uniform distribution is as $p(k_i|q_i) = 1/L_k$. So, the Kullback–Leibler (KL) divergence for measuring the probability distribution between can be expressed as Equation (10).

$$KL(q\|p) = \ln \sum_{i=1}^{L_K} e^{q_i, k_j^T / \sqrt{d}} - \frac{1}{L_x} \sum_{i=1}^{L_K} q_i, k_j^T / \sqrt{d} - \ln L_K \tag{10}$$

Removing the constant term [29], we define the $i$-th query's sparsity measurement as expressed by Equation (11).

$$M(q_i, K) = \ln \sum_{i=1}^{L_K} e^{q_i, k_j^T / \sqrt{d}} - \frac{1}{L} \sum_{i=1}^{L_K} q_i, k_j^T / \sqrt{d} \tag{11}$$

The dimension of $\overline{Q}$ is consistent with $Q$. We sort the values computed based on M and select the top 10% to fill in $\overline{Q}$. The remaining values are filled with the overall mean 10%, which was obtained through testing, and the specific results are shown in Section 4.2. So, the prob_attention can be expressed as Equation (12).

$$A(Q, K, V) = softmax(\frac{\overline{Q}K^T}{\sqrt{d}})V \tag{12}$$

Conv1d is a one-dimensional convolutional module with a kernel size of 3 and a stride of 2. Its main role is to reduce the size of the feature map and lower the computational complexity. Finally, the data is normalized through layer norm as in Equation (13).

$$\text{LayerNorm}(x) = \gamma \cdot \frac{x - \mu}{\sigma} + \beta \tag{13}$$

where $\mu = \frac{1}{L_x} \sum_{i=1}^{L_x} x_i$, $\sigma = \sqrt{\frac{1}{L_x} \sum_{i=1}^{L_x} (x_i - \mu)^2}$, $\gamma$ and $\beta$ are learnable scaling and bias parameters. So output can be expressed as Equation (14).

$$\begin{aligned} \text{Output}^E = \text{LayerNorm}\Big(A\Big(ELU\Big(\text{Conv1d}\Big(A\Big(\text{INPUT}^E\Big) + \text{INPUT}^E\Big)\Big)\Big) + \\ ELU\Big(\text{Conv1d}\Big(A\Big(\text{INPUT}^E\Big) + \text{INPUT}^E\Big)\Big)\Big) \end{aligned} \tag{14}$$

where $INPUT^E$ is the input after undergoing Encoder embedding, and $OUTPUT^E$ is the output of the Encoder.

Self-attention computation requires memory of $O(L_Q L_K)$ and comes at the cost of quadratic dot product calculations. In this paper, because M values are used for calculation, the overhead of M is $O(L_Q L_K)$, and the computational cost of attention calculation after M screening is $O(L_Q L_K / 10)$. Therefore, the overall cost is higher with $O(L_Q L_K / 10)$ compared to regular self-attention.

We have implemented the prob attention in Python 3.6 with Pytorch 1.8.0. The pseudo-code is given in Algorithm 1.

---

**Algorithm 1:** Prob attention

---

1 set hyperparameter u, u = 10%
2 set the sample score $S = \frac{QK^T}{\sqrt{d}}$ by row
3 compute the measurement $M = \ln(S) - mean(S)$
4 set Top u queries under M as $\overline{Q}$
5 set $S_1 = softmax(\overline{Q}K^T / \sqrt{d})V$
6 set $S_0 = mean(V)$
7 set $S = \{S_1, S_0\}$ by their original rows accordingly
   **Result:** feature map S

---

*3.3. Decoder*

The Decoder block is similar to the Encoder block, but with an additional Multi-head Attention layer. The Decoder input vector contains additional $Z^t \in \mathbb{R}^{L_k \times d_{model}}$ placeholders for prediction, where *k* is the number of errors to be predicted. The values at these positions are filled with zeros and can be expressed as in Equation (15).

$$INPUT^D = Concat\left(\overline{F^t} + \overline{G^t} + Z^t\right) \in \mathbb{R}^{(L_x + L_x + L_k) \times d_{model}} \tag{15}$$

where $\overline{F^t} \in \mathbb{R}^{L_x \times d_{model}}, \overline{G^t} \in \mathbb{R}^{L_x \times d_{model}}$. The Decoder input vectors serve as the input vectors for the neural network. AGEE as depicted in Figure 3.

A standalone attention module is incapable of providing a comprehensive global representation of temporal relationships in sequence data. Hence, by partitioning $d_{model}$ into multiple segments and independently computing them using attention modules, the outcomes are concatenated along the relevant dimension. This methodology is termed as Multi-head Attention. The computation for the *i*-th attention module can be articulated through Equation (16).

$$A(Q_i, K_i, V_i) = soft\max\left(\frac{Q_i K_i^T}{\sqrt{d}}\right)V_i \tag{16}$$

Therefore, Multi-head Attention can be represented as Equation (17).

$$Multihead(X) = (Concat(A(Q_1, K_1, V_1), \dots, A(Q_h, K_h, V_h)))W^o \tag{17}$$

where attention represents the calculation of the attention mechanism, Concat denotes the concatenation of outputs from multiple attention heads along the last imension, and $W^o$ is the weight matrix for the linear transformation of the concatenated output.

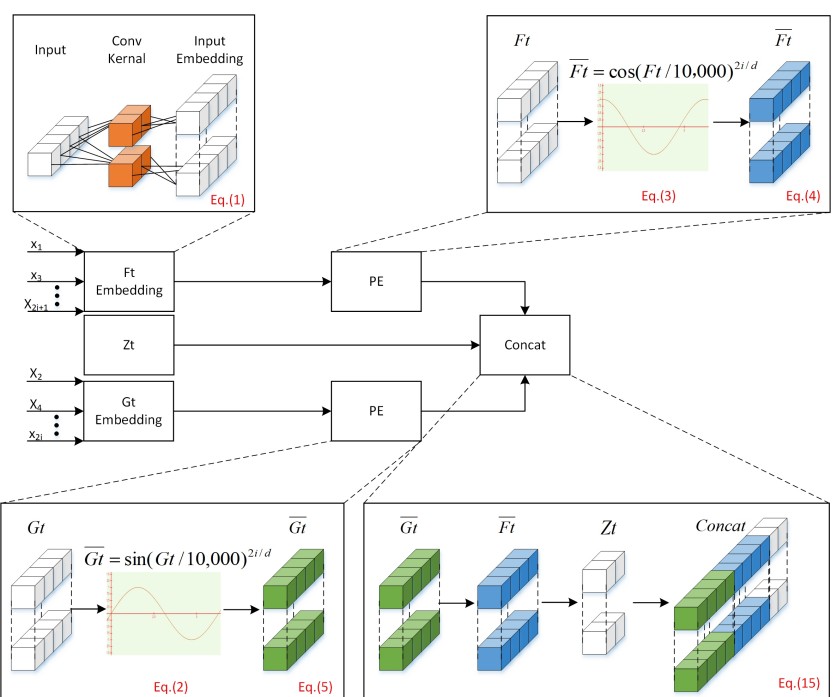

**Figure 3.** The input to the Decoder, after undergoing the AGEE, includes an additional placeholder, $Z^t$, at the output position, filled with zeros.

## 4. Experimentation

### 4.1. Data Collection

To validate the effectiveness of PAGEformer, we collaborated with North Navigation Control Technology Co., Ltd. and collected measurement data during the manufacturing process of folding rudder blades for unmanned aerial vehicles produced by the company. The Mechanical Assembly Relationship Dataset (MARD) comprises 2262 sets of data, with each set consisting of 13 measurement data items and 1 assembly error data item (Table 1).

To further evaluate the performance of PAGEformer, additional tests were conducted using the following three publicly available datasets:

(1) ETT (Electricity Transformer Temperature): This dataset includes two categories of data collected at 1 h frequency (ETTh) and 15 min frequency (ETTm), each containing 7 items of feature data.
(2) ECL (Electricity Consumption Load): This dataset contains electricity consumption data of 321 customers, with each record containing 320 items of feature data.
(3) Weather: This dataset contains climate data for nearly 1600 regions in the United States, with data collected at an hourly frequency. Each record includes 12 items of feature data.

**Table 1.** Mechanical assembly relationship dataset (partial).

| Left Shaft Hole | Groove Surface | Locking Block Groove | Right Shaft Hole | Left | Front | Flat | Height of the Hole Center | Lock Block Left | Lock Block Right | Right | Behind | Hole Position Height | Shaking Amount |
|---|---|---|---|---|---|---|---|---|---|---|---|---|---|
| 4.48 | 6 | 4.52 | 4.55 | 5.975 | 4.51 | 5.96 | 9.17 | 4.51 | 4.51 | 5.995 | 4.54 | 9.595 | 0.7 |
| 4.51 | 5.975 | 4.52 | 4.56 | 5.98 | 4.54 | 6.015 | 9.215 | 4.5 | 4.52 | 5.96 | 4.54 | 9.52 | 0.8 |
| 4.52 | 5.98 | 4.52 | 4.55 | 6.005 | 4.49 | 5.99 | 9.29 | 4.45 | 4.48 | 5.955 | 4.48 | 9.495 | 1.4 |
| 4.53 | 6.055 | 4.52 | 4.56 | 5.99 | 4.54 | 6.095 | 9.255 | 4.5 | 4.51 | 5.96 | 4.53 | 9.5 | 0.8 |
| … | … | … | … | … | … | … | … | … | … | … | … | … | … |

We employed single 4090 GPU for training, and the training of MARD took approximately 10 h. PAGEformer can adapt input data based on the required measurement data

for components, such as data for 20 measurements and two predicted errors. Thus, it can predict various fitting errors of components in industrial production. However, due to the large amount of measurement data for components, it is necessary to select relevant data that play a decisive role in dimensional fitting, which typically requires collaboration with mechanical engineers. Regarding the scalability to other domains, this paper demonstrates PAGEformer's predictive performance on long time-series data, as shown in Section 4.2, which proves its effectiveness in predicting data with long time-series correlations for the majority of cases.

To facilitate the introduction of hyperparameters, the following parameters are based on the data in MARD. The training data has a batch size of 32. The Encoder sequence length is 13, which is the dimensionality parameter used in MARD for predicting the final error. The Decoder sequence length is 14, comprising 13 dimensionality datums and 1 placeholder datum filled with 0. Conv1D has a kernel size of 3, a stride of 1, padding of 1, and outputs a dimensionality of 512 for embedding purposes. In prob attention, the threshold for filtering $\overline{Q}$ is set to 0.1. LayerNorm is used to prevent division by zero, thus adding 0.00001 to the denominator. The Elu activation function is chosen with an alpha value of 1. Dropout is set to 0.5. In the Decoder, the number of heads in multihead attention is eight. The Adam optimizer is chosen with a learning rate of 0.0005. MSE is selected as the loss function. The FC output dimensionality is 14, with the first 13 representing the input data and the last one representing the predicted data output by the model.

*4.2. Experimental Results and Discussion*

The performance of the PAGEformer method can be assessed through MARD. The evaluation used two metrics: Mean Absolute Error (MAE) and Mean Squared Error (MSE). The test results of different values of $\overline{Q}$ in MARD are displayed in Table 2 and the line graph is shown in Figure 4.

**Table 2.** The test results of MARD with different $\overline{Q}$ values.

| Top X% to Fill | MSE | MAE |
| --- | --- | --- |
| 10% | 0.0375 | 0.1565 |
| 20% | 0.0783 | 0.2293 |
| 30% | 0.0723 | 0.2086 |
| 40% | 0.0700 | 0.2047 |
| 50% | 0.1149 | 0.2969 |
| 60% | 0.0799 | 0.2233 |
| 70% | 0.0490 | 0.1649 |
| 80% | 0.0974 | 0.2516 |
| 100% | 0.0850 | 0.2264 |

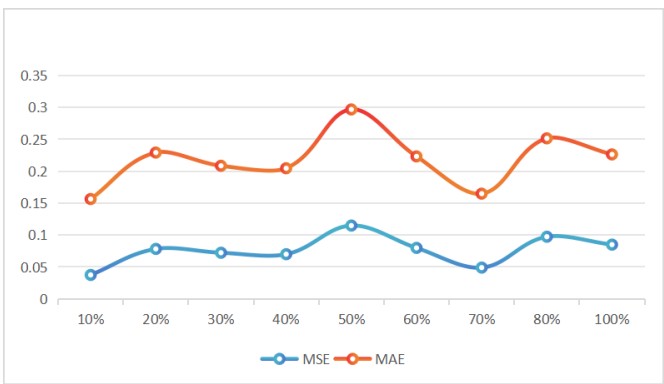

**Figure 4.** A line graph of the MSE for different values of $\overline{Q}$.

Table 3 provides a comparison of the MSE and MAE metric values for PAGEformer, Reformer, and ARIMA on the MARD. The data demonstrates that the accuracy of PAGEformer on MARD has increased by 15.3% compared to the best-performing ARIMA.

**Table 3.** Comparative test results of various methods for MARD.

| Method | Metric | Value |
|--------|--------|-------|
| PAGEformer | MSE | 0.0375 |
|  | MAE | 0.1565 |
| Reformer | MSE | 0.0492 |
|  | MAE | 0.1701 |
| ARIMA | MSE | 0.0456 |
|  | MAE | 0.1833 |

In order to further evaluate the performance of PAGEformer, we conducted comparative tests with seven different methods using three publicly available datasets. The seven existing methods employed in the comparison are as follows:

(1) Informer, (2) LogTrans, (3) Reformer, (4) LSTMa, (5) DeepAR, (6) ARIMA, (7) Prophet.

In this case as well, the individual methods were evaluated using the MSE and MAE metrics. All models were trained and tested on a single Nvidia 4090 GPU. The experimental results obtained are presented in Table 4. In order to show the comparison results more intuitively, Figure 5 is drawn according to the data in Table 4.

**Table 4.** A comparison of PAGEformer with mainstream methods.

| Method | | PAGEformer | | Informer | | LogTrans | | Reformer | | LSTMa | | DeepAR | | ARIMA | | Prophet | |
|--------|------------|-----|-----|-----|-----|-----|-----|-----|-----|-----|-----|-----|-----|-----|-----|-----|-----|
| Metric | Input Length | MSE | MAE | MSE | MAE | MSE | MAE | MSE | MAE | MSE | MAE | MSE | MAE | MSE | MAE | MSE | MAE |
| ETTh1 | 24 | 0.082 | 0.225 | 0.092 | 0.246 | 0.103 | 0.259 | 0.222 | 0.389 | 0.114 | 0.272 | 0.107 | 0.280 | 0.108 | 0.284 | 0.115 | 0.275 |
|  | 48 | 0.119 | 0.274 | 0.161 | 0.322 | 0.167 | 0.328 | 0.284 | 0.445 | 0.193 | 0.358 | 0.162 | 0.327 | 0.175 | 0.424 | 0.168 | 0.330 |
|  | 168 | 0.186 | 0.358 | 0.187 | 0.355 | 0.207 | 0.375 | 1.522 | 1.191 | 0.236 | 0.392 | 0.239 | 0.422 | 0.396 | 0.504 | 1.224 | 0.763 |
|  | 336 | 0.182 | 0.350 | 0.215 | 0.369 | 0.230 | 0.398 | 1.860 | 1.124 | 0.590 | 0.698 | 0.445 | 0.552 | 0.468 | 0.593 | 1.549 | 1.820 |
|  | 720 | 0.218 | 0.325 | 0.257 | 0.421 | 0.273 | 0.463 | 2.112 | 1.436 | 0.683 | 0.768 | 0.658 | 0.707 | 0.659 | 0.766 | 2.735 | 3.253 |
| ETTh2 | 24 | 0.090 | 0.229 | 0.099 | 0.241 | 0.102 | 0.255 | 0.263 | 0.437 | 0.155 | 0.307 | 0.098 | 0.263 | 3.554 | 0.445 | 0.199 | 0.381 |
|  | 48 | 0.147 | 0.301 | 0.159 | 0.317 | 0.169 | 0.348 | 0.458 | 0.545 | 0.190 | 0.348 | 0.163 | 0.341 | 3.190 | 0.474 | 0.304 | 0.462 |
|  | 168 | 0.263 | 0.415 | 0.235 | 0.390 | 0.246 | 0.422 | 1.029 | 0.879 | 0.385 | 0.514 | 0.255 | 0.414 | 2.800 | 0.595 | 2.145 | 1.068 |
|  | 336 | 0.293 | 0.439 | 0.258 | 0.423 | 0.267 | 0.437 | 1.668 | 1.228 | 0.558 | 0.606 | 0.604 | 0.607 | 2.753 | 0.738 | 2.096 | 2.543 |
|  | 720 | 0.295 | 0.439 | 0.285 | 0.442 | 0.303 | 0.493 | 2.030 | 1.721 | 0.640 | 0.681 | 0.429 | 0.580 | 2.878 | 1.044 | 3.355 | 4.664 |
| ETTm1 | 24 | 0.034 | 0.147 | 0.034 | 0.160 | 0.065 | 0.202 | 0.095 | 0.228 | 0.121 | 0.233 | 0.091 | 0.243 | 0.090 | 0.206 | 0.120 | 0.290 |
|  | 48 | 0.063 | 0.195 | 0.066 | 0.194 | 0.078 | 0.220 | 0.249 | 0.390 | 0.305 | 0.411 | 0.219 | 0.362 | 0.179 | 0.306 | 0.133 | 0.305 |
|  | 96 | 0.193 | 0.365 | 0.187 | 0.384 | 0.199 | 0.386 | 0.920 | 0.767 | 0.287 | 0.420 | 0.364 | 0.496 | 0.272 | 0.399 | 0.194 | 0.396 |
|  | 288 | 0.398 | 0.546 | 0.409 | 0.548 | 0.411 | 0.572 | 1.108 | 1.245 | 0.524 | 0.584 | 0.948 | 0.795 | 0.462 | 0.558 | 0.452 | 0.574 |
|  | 672 | 0.529 | 0.643 | 0.519 | 0.665 | 0.598 | 0.702 | 1.793 | 1.528 | 1.064 | 0.873 | 2.437 | 1.352 | 0.639 | 0.697 | 2.747 | 1.174 |
| weather | 24 | 0.109 | 0.236 | 0.119 | 0.256 | 0.136 | 0.279 | 0.231 | 0.401 | 0.131 | 0.254 | 0.128 | 0.274 | 0.219 | 0.355 | 0.302 | 0.433 |
|  | 48 | 0.181 | 0.313 | 0.185 | 0.316 | 0.206 | 0.356 | 0.328 | 0.423 | 0.190 | 0.334 | 0.203 | 0.353 | 0.273 | 0.409 | 0.445 | 0.536 |
|  | 168 | 0.259 | 0.377 | 0.269 | 0.404 | 0.309 | 0.439 | 0.654 | 0.634 | 0.341 | 0.448 | 0.293 | 0.451 | 0.503 | 0.599 | 2.441 | 1.142 |
|  | 336 | 0.292 | 0.397 | 0.310 | 0.422 | 0.359 | 0.484 | 1.792 | 1.093 | 0.456 | 0.554 | 0.585 | 0.644 | 0.728 | 0.730 | 1.987 | 2.468 |
|  | 720 | 0.299 | 0.425 | 0.361 | 0.471 | 0.388 | 0.499 | 2.087 | 1.534 | 0.866 | 0.809 | 0.499 | 0.596 | 1.062 | 0.943 | 3.859 | 1.144 |
| ECL | 48 | 0.261 | 0.363 | 0.238 | 0.368 | 0.280 | 0.429 | 0.971 | 0.884 | 0.493 | 0.539 | 0.204 | 0.357 | 0.879 | 0.764 | 0.524 | 0.595 |
|  | 168 | 0.360 | 0.426 | 0.442 | 0.514 | 0.454 | 0.529 | 1.671 | 1.587 | 0.723 | 0.655 | 0.315 | 0.436 | 1.032 | 0.833 | 2.725 | 1.273 |
|  | 336 | 0.432 | 0.464 | 0.501 | 0.552 | 0.514 | 0.563 | 3.528 | 2.196 | 1.212 | 0.898 | 0.414 | 0.519 | 1.136 | 0.876 | 2.246 | 3.077 |
|  | 720 | 0.423 | 0.474 | 0.543 | 0.578 | 0.558 | 0.609 | 4.891 | 4.047 | 1.511 | 0.966 | 0.563 | 0.595 | 1.251 | 0.933 | 4.243 | 1.415 |
|  | 960 | 0.537 | 0.540 | 0.594 | 0.638 | 0.624 | 0.645 | 7.019 | 5.105 | 1.545 | 1.006 | 0.657 | 0.683 | 1.370 | 0.982 | 6.901 | 4.264 |

Table 4 illustrates that PAGEformer significantly enhances the inference ability across all datasets. Compared to the best model, PAGEformer accuracy has improved by 15.17%, 17.17%, and 9.5% on the publicly available ETH, Weather, and ECL datasets, respectively. The aforementioned data substantiate that the proposed method can effectively measure hidden correlations in data, thus predicting the overall errors of the assembly process. Additionally, in Table 4, PAGEformer demonstrated accuracy in predicting sequence data, indicating its capability to learn the correlation of different errors and perform well in predicting time series. This suggests that the prediction ability of PAGEformer has good

scalability. To further validate the improvements in PAGEformer, we conducted ablation experiments with two modifications: gap and filling encoding and prob attention. The experimental results are presented in Table 5.

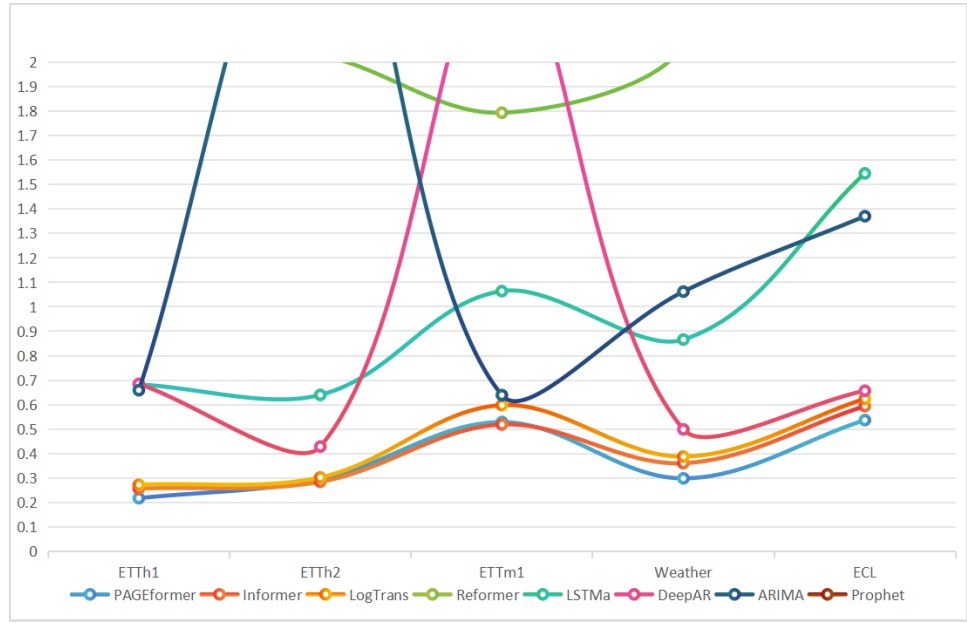

**Figure 5.** Comparison of PAGEformer with other methods on mainstream datasets.

**Table 5.** A comparison of ablation experiments for PAGEformer with MARD.

| Method | Metric | Value |
|---|---|---|
| Prob and AGEE | MSE | 0.0375 |
| | MAE | 0.1565 |
| removes Prob | MSE | 0.0453 |
| | MAE | 0.1642 |
| removes AGEE | MSE | 0.0385 |
| | MAE | 0.1599 |
| removes Prob and AGEE | MSE | 0.1847 |
| | MAE | 0.3688 |

## 5. Conclusions

This study explores the prediction of Assembly Geometric Errors and introduces PAGEformer, a neural network capable of determining the feature distance of assembly relationships using collected component data. By integrating an enhanced and efficient long temporal attention structure, it improves the prediction of errors post-assembly. To validate these improvements, the study collects a substantial amount of component data and post-assembly errors from an actual manufacturing environment, creating the Mechanical Assembly Relationship Dataset (MARD). Experimental results show that PAGEformer achieves an accuracy on MARD 15.3% higher than that of ARIMA. To further evaluate PAGEformer performance on extended temporal data, the study conducts tests on public datasets and performs ablation experiments to scrutinize the effectiveness of the enhancements. The outcomes on public datasets showcase PAGEformer's commendable performance in standard long temporal tests, and the ablation experiments affirm the effectiveness of the two proposed enhancements.

**Author Contributions:** Software, P.L. and J.S.; Writing—original draft, W.W.; Writing—review and editing, B.W.; Project administration, H.L. and B.N. All authors have read and agreed to the published version of the manuscript.

**Funding:** This work was supported by the National Natural Science Foundation of China (62303070) and the Key Research Project of Science and Technology Department of Jilin Province (20210201113GX).

**Data Availability Statement:** The data presented in this study are available on request from the corresponding author (Due to the sensitive nature of the MARD data involving military confidentiality, permission needs to be obtained. The remaining data are from public datasets).

**Conflicts of Interest:** Wu Wang, Botong Niu and Jing Sun were employed by North Navigation Control Technology Co., Ltd., and they declare no conflict of interest. The funders had no role in the design of the study; in the collection, analyses, or interpretation of data; in the writing of the manuscript; or in the decision to publish the results. The authors declare no conflict of interest.

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
