# Peer review of "Predicting Assembly Geometric Errors Based on Transformer Neural Networks"

_machines, doi:10.3390/machines12030161_

Round 1

Reviewer 1 Report

Comments and Suggestions for Authors

In this work, the authors introduced PAGEformer, a Transformer-based neural network model tailored for predicting assembly geometric errors in manufacturing. PAGEformer integrates enhanced self-attention mechanisms and positional information generation to effectively capture long-range assembly relationships. They collected real assembly data for UAV rudder blades, establishing a Mechanical Assembly Relationship Dataset (MARD), and conducted extensive testing, showing significant accuracy improvements over existing models across various datasets. This approach offers a promising avenue for improving product quality without substantial cost increases in manufacturing processes. The contributions of the work are significant and can be accepted after addressing some critical concerns.

1. The Introduction section is written tangentially. The authors are suggested to expand the literature review [5-10] and [11-16] to improve the knowledge base of the topic. Moreover, the authors should write the background of different transformer models used in the prediction of different datasets, along with their limitations. Then, based on the same, please mention the novelty of the proposed transformer.

2. In Figure 2, the encoder output is fed into the multi-attention decoder block and then to the FC which is similar to the general transformer models The authors are suggested to provide more details on the specific enhancements made to the self-attention mechanism in PAGEformer and how they contribute to handling long-range dependencies more effectively.

3. Were there any noteworthy findings or insights gained from the ablation experiments conducted in the study? How do these findings contribute to our understanding of the effectiveness of different components of PAGEformer?

4. The authors are suggested to provide insights into the computational requirements and scalability of PAGEformer.

5. How does the model's efficiency and resource utilization compare to other approaches, particularly in scenarios involving large-scale manufacturing datasets?

6. The authors are suggested to use graphical representations to show the numerical results and findings to improve the readability of the manuscript.

7. What considerations were taken into account when selecting the datasets for testing PAGEformer's performance? Were there any characteristics or features of these datasets that influenced their suitability for evaluation?

Reviewer 2 Report

Comments and Suggestions for Authors

Please view the attachment.

Comments on the Quality of English Language

English is fine in general. Need to check for typo and minor grammar errors.

Round 2

Reviewer 1 Report

Comments and Suggestions for Authors

The authors have addressed most of the concerns raised by the reviewer. However, it would be nice if the authors provided more information on training the proposed transformer architecture. It is unclear which and how training (hyper)parameters lead to a total training time of ~10 hours. There should be clear information on training and architecture parameters. 
